# Associations between hematologic dynamics during pregnancy and obstetric complications: A retrospective observational study

Veronica Tozzo[1,2,3☺*], Rachel Petherbridge[1,2☺], Kaitlyn James[4], Sarah Hsu[5,6], Deepti Pant[7], Chloe Michalopoulos[5], Brody H. Foy[1,2,8], Tanayott Thaweethai[7,9], Christopher Mow[1,10], Jacqueline Maya[5,9], Carolina Batlle Camero[11], Lydia Shook[4], Kathryn J. Gray[12], Logan Mauney[4], John M. Higgins[1,2‡*], Camille E. Powe[4,5,6,9‡*]

1 Department of Pathology and Center for Systems Biology, Massachusetts General Hospital, Boston, Massachusetts, United States of America, 2 Department of Systems Biology, Harvard Medical School, Boston, Massachusetts, United States of America, 3 Center for Neurobehavioral Genetics, Semel Institute for Neuroscience and Human Behavior, University of California Los Angeles, Los Angeles, California, United States of America, 4 Department of Obstetrics and Gynecology, Massachusetts General Hospital, Boston, Massachusetts, United States of America, 5 Diabetes Unit, Department of Medicine, Massachusetts General Hospital, Boston, Massachusetts, United States of America, 6 Broad Institute, Cambridge, Massachusetts, United States of America, 7 Biostatistics Center, Division of Clinical Research, Massachusetts General Hospital, Boston, Massachusetts, United States of America, 8 Department of Laboratory Medicine & Pathology, University of Washington, Seattle, Washington, United States of America, 9 Harvard Medical School, Boston, Massachusetts, United States of America, 10 Mass General Brigham Enterprise Research IS, Boston, Massachusetts, United States of America, 11 University of Puerto Rico, School of Medicine, San Juan, Puerto Rico, 12 Department of Obstetrics and Gynecology, University of Washington School of Medicine, Seattle, Washington, United States of America

☺ These authors contributed equally to this work.
‡ JMH and CEP also contributed equally to this work.
* vtozzo@mednet.ucla.edu (VT); higgins.john@mgh.harvard.edu (JMH); camille.powe@mgh.harvard.edu (CEP)

## Abstract

### Background

Pregnancy alters hematologic state as measured by complete blood count (CBC), but the longitudinal changes in CBC indices that define healthy pregnancies are not well established. In a large cohort based at an academic health system in the United States, we aimed to define reference intervals and typical longitudinal changes in CBC indices during pregnancy. We then tested for associations between extreme CBC values for gestational age or extreme longitudinal changes in CBC indices and obstetric complications.

### Methods and findings

We studied nine CBC indices in individuals with singleton pregnancies who delivered after 30 weeks' gestation and presented for prenatal care prior to 20 weeks. The

**Data availability statement:** All aggregated data analyzed and discussed is provided in the manuscript or its published supplementary materials. The individual patient data used in this study is from Mass General Brigham (MGB) patients and cannot be shared publicly because of restrictions related to patient privacy and confidentiality. Data can be shared in accordance with the Data Sharing Policy of MGB which requires a Data Use Agreement prior to any exchange of human subject data with an external party for research purposes. Data are available from the MGB Institutional Review Board (contact via email, IRB@mgb.org, or phone +1-857-282-1900) for researchers who meet these criteria for access.

**Funding:** V.T, R.P., and J.M.H were supported by NIH DK123330. V.T, R.P, J.M.H, and C.E.P were supported by NIH R01HD104756. J.M.H was supported by the Gates Foundation Award 090035. K.J.G reports funding from NIH/NHLBI K08 HL146963. L.S. reports funding from NIH/NICHD 5K12HD103096-04. C.E.P. was supported by NIH/NIDDK K26DK138346. J.M. was supported by an NIH T32 training grant for Endocrinology (T32DK007028-47S1). Further details are available at reporter.nih.gov. The MGH Maternal Health Cohort was supported by the MGH Claflin Distinguished Scholar Award and the MGH Physician-Scientist Development Award (to C.E.P., details at the Mass General Research Institute site). The funders had no role in study design, data collection and analysis, decision to publish, or preparation of the manuscript.

**Competing interests:** I have read the journal's policy and the authors of this manuscript have the following competing interests: K.J.G. reports consulting for BillionToOne, Aetion, and Janssen Global, Inc outside the scope of the submitted work. C.E.P. has received fees and royalties from Mediflix and UpToDate (Wolters Kluwer), respectively, for presentations and articles related to diabetes over which she had full control of content. C.E.P. and L.L.S. receive research support from Dexcom through their institution for an unrelated project. All other authors have no competing interests to declare.

**Abbreviations:** ACOG, American Congress of Obstetricians and Gynecologists; BMI, body mass index; CBC, complete blood count; EHR,

electronic health record (EHR)-based Maternal Health Cohort (Massachusetts General Hospital; 1998–2016) formed our discovery cohort of 45,992 pregnancies, 18% of which had relevant complications. We developed a validation cohort of 48,868, 27% with complications from EHR data in the Mass General Brigham healthcare system from 2016 to 2024. In pregnancies without complications in the discovery cohort, we derived gestational-age-specific reference intervals (2.5th–97.5th percentile) and established typical intra-pregnancy longitudinal changes. In the validation cohort, we then tested CBC values outside of the 26–29 weeks' gestation reference interval and CBC rare changes (uncommon changes in magnitude and direction) between 7–14 and 26–29 weeks' gestation for association with a composite outcome (hypertensive disorders of pregnancy, small for gestational age birthweight, preterm birth) and its individual components using generalized estimating equations. Derived reference intervals differed from those in the literature for mean red cell volume, mean red cell hemoglobin, red cell count, and mean red cell hemoglobin concentration; reference intervals for other indices were similar to those previously published. In validation, hematocrit, hemoglobin, and red cell count values above their gestational-age specific reference intervals were associated with increased risk of the composite obstetric outcome: odds ratios (ORs) of 1.4 (95% CI [1.2, 1.5] $p < 0.0001$) for hematocrit; 1.6 (95% CI [1.4, 1.8] $p < 0.0001$) for hemoglobin; and 1.6 (95% CI [1.4, 1.7] $p < 0.0001$) for red cell count. Uncommon increases in hemoglobin (>0.67 g/dL) or red cell count (>0.07 $10^6$/mm$^3$) between 7–14 weeks' and 26–29 weeks' gestation were associated with increased risk for preterm birth, OR for hemoglobin 1.9 (95% CI [1.5, 2.5] $p < 0.0001$) and red cell count 2.1 (95% CI [1.7, 2.6] $p < 0.0001$). Limitations include the retrospective nature of the study and the exclusion of pregnancies without prenatal care prior to 20 weeks' and pregnancies delivered before 29 weeks' gestation.

## Conclusions

Elevated red blood cell-related measurements and unusually large intra-pregnancy increases in those measures are associated with subsequent obstetric complications.

## Author summary

### Why was this study done?

- Pregnancy induces a host of physiologic changes, including changes in the size and composition of circulating blood cell populations.

- Complete blood count tests are commonly used in prenatal care to look for conditions like anemia and infection, but it is not well understood which pregnancy-related changes are normal and which might signal a higher risk of pregnancy complications.

electronic health record; GEE, generalized estimating equations; HCT, hematocrit; HDP, hypertensive disorders of pregnancy; HGB, hemoglobin; ICD, International Classification of Disease; MCH, mean red cell hemoglobin content; MCHC, mean red cell hemoglobin concentration; MCV, mean red cell volume; MGB, Mass General Brigham; MGB IRB, Mass General Brigham Institutional Review Board; OR, odds ratios; PLT, platelet count; PPV, positive predictive value; RBC, red blood cell count; RDW, red cell distribution width; RECORD, Reporting of Studies Conducted using Observational Routinely-Collected Data; SGA, small for gestational age birth weight; WBC, white blood cell count.

- Understanding these patterns could help identify pregnancies at higher risk using tests that are already part of standard care.

### What did the researchers do and find?

- We studied routine complete blood count test results from over 94,000 pregnancies receiving care in a large U.S. health system.

- We described normal complete blood count test values and typical changes during pregnancy in people who did not develop complications.

- We showed that higher-than-expected red blood cell-related measurements between 26 and 29 weeks' gestation, as well as increases in these measurements compared to earlier in pregnancy, were linked to a higher risk of complications such as preterm birth and high blood pressure.

### What do these findings mean?

- Simple blood tests that are already used in prenatal care can provide early clues about the risk of pregnancy complications.

- Looking at how blood test results change over time may help identify risk even when individual test results appear "normal."

- This approach could support earlier and more personalized care and may be especially useful in settings where access to specialized testing is limited.

- Limitations included the retrospective nature of the study and the exclusion of both pregnancies without prenatal care prior to 20 weeks and pregnancies delivered before 29 weeks' gestation.

### Introduction

Pregnancy leads to alterations in the hematologic system [1,2] that are reflected in changes to complete blood counts (CBCs) routinely measured during prenatal care [3,4], but consensus is lacking on what constitutes normal physiological CBC change in pregnancy [5,6]. Outside pregnancy, abnormal CBC values and longitudinal changes in these values have been shown to be useful for clinical risk stratification [7], but in pregnancy, they have not been comprehensively evaluated as risk markers [8–10]. With ~85% of pregnant individuals in the United States (US) receiving longitudinal prenatal care that routinely includes CBC assessments [11], there is a potential missed opportunity for early identification of obstetric complications.

Pregnancy-specific reference intervals are available but are derived from cross-sectional studies [12–17] and meta-analyses of studies with small to moderate sample sizes [3,4,16,18,19]. Utilization of pregnancy-specific reference intervals by clinicians is limited, and their clinical utility has not been definitively demonstrated [3,4,6] with the exception of pregnancy-specific lower limits of hemoglobin used to diagnose anemia [20]. Intra-pregnancy changes in CBCs have not been systematically

investigated, with analysis limited to small studies [15,18,19,21,22] or cross-sectional electronic health record (EHR) analysis [5,17,23], and there has been no evaluation of intra-pregnancy CBC changes for routine risk stratification of pregnancy.

Here, we derive gestational age-specific reference intervals for CBC indices and their intra-pregnancy longitudinal changes in a large US-based cohort. We then use these reference intervals and longitudinal changes to systematically evaluate the associations of extreme values and rare dynamics with pregnancy complications.

## Methods

Study methods are summarized below; additional details appear in the Supplementary Methods in S1 Appendix. This study did not rely on a prospective protocol or analysis plan, and it is reported as per the Reporting of Studies Conducted using Observational Routinely-Collected Data (RECORD) guideline (S1 Checklist).

### Study setting and cohort description

Our discovery cohort was the EHR-based Maternal Health Cohort, which includes all pregnant patients who received prenatal care or had a delivery at Massachusetts General Hospital between 1998 and 2016 [24–26]. We developed an out-of-sample validation cohort of pregnancies delivered across the Mass General Brigham (MGB) health system between 2016 and 2024. In both discovery and validation cohorts, we included singleton pregnancies ending in live birth and excluded those with known blood-related disorders (including pre-existing diagnosis of immune thrombocytopenia purpura, hemolytic anemia, or a genetic blood disorder based on available diagnoses in the problem list or International Classification of Disease (ICD) 9/10 codes, see Text A in S1 Appendix for details), with known chronic hypertension or elevated blood pressure prior to 20 weeks' gestation (see Text A in S1 Appendix for details), who delivered prior to 29 weeks' gestation (required for determining exposure), or who were missing data on birth weight or neonatal sex (required for determining a primary outcome). In the discovery cohort, we additionally excluded pregnancies with transfusions of plasma, platelets, or red blood cells during or within 8 weeks prior to pregnancy, and those without a prenatal visit prior to 20 weeks' gestation (both required for primary outcome ascertainment in this cohort), and we required at least one CBC checked in pregnancy. In the validation cohort, we required a CBC both at 7–14 weeks' and 26–29 weeks' gestation (required for exposure ascertainment).

### CBC indices

We considered the following CBC indices: hematocrit (HCT), hemoglobin (HGB), white blood cell count (WBC), red blood cell count (RBC), platelet count (PLT), mean red cell volume (MCV), mean red cell hemoglobin content (MCH), red cell distribution width (RDW), and mean red cell hemoglobin concentration (MCHC). See Text B in S1 Appendix for detail on measuring instruments. After conducting a systematic evaluation of CBC parameter stability, we excluded mean platelet volume due to shifts across hematological analyzers (Fig A in S1 Appendix). Based on the timing of routine CBC screening (Fig B in S1 Appendix), we considered three pregnancy timepoints: 7–14 weeks' gestation (7 weeks 0 days to 14 weeks 0 days), 26–29 weeks' gestation (26 weeks 0 days to 29 weeks 0 days), and pre-delivery, which was pregnancy-specific and defined as the CBC closest to delivery within a 0–7 days window prior to delivery. We also considered pre-pregnancy CBCs for individuals who had ≤2 CBCs at least 6 months apart in the previous five years with no ICD codes indicating an illness or pregnancy at the time of the CBC (see Text C in S1 Appendix). When multiple CBCs were present in any of the considered windows, we included the first chronological CBC for analysis.

### Reference intervals and longitudinal dynamics in pregnancies without complications

Reference intervals are defined by studying healthy populations, with inclusion and exclusion criteria determined by the specific diagnostic and demographic scenario [27]. Therefore, we defined reference intervals for pregnancy based on

distributions of test results obtained for pregnancies that delivered at term (≥37 weeks' gestation) without complications of interest (preterm birth, small for gestational age birthweight, or hypertensive disorders of pregnancy, see Associations with adverse outcomes section below) in the discovery cohort. We defined gestational age-specific reference intervals as the 2.5th and 97.5th percentiles for each CBC index at each time point, using all available CBCs at that time point. Literature reference intervals were collected from two commonly used online sources [28,29] derived from meta-analyses [4]. In sensitivity analyses, we determined gestational age-specific intervals in pregnancies without anemia [20,30] (HGB < 11 g/dL at pre-pregnancy, 7–14 week's gestation and prior to delivery, and HGB < 10.5 g/dL at 26–29 weeks' gestation), in pregnancies with intravenous or oral iron, and in the subset of term pregnancies without prior chronic conditions or pregnancy-related issues recorded in the EHR.

We investigated intra-pregnancy longitudinal changes in the discovery cohort by analyzing a subset of term pregnancies without complications for which CBCs were available prior to pregnancy and at three examined pregnancy time points (7–14 weeks' gestation, 26–29 weeks' gestation, and pre-delivery). The sample size for this analysis was primarily limited by the requirement for a pre-pregnancy CBC, as not everyone in the cohort received pre-pregnancy care in the health system. In a sensitivity analysis, we included all pregnancies with at least two consecutive time points. To evaluate typical longitudinal changes in CBC indices across pregnancy, we fit linear mixed-effects models adjusted for age and 1st trimester body mass index (BMI) [26] (see Text D in S1 Appendix for BMI calculation), adding random effects for individual pregnancy, individual participant, parity, delivery year, and prenatal care site; more details are available in Text E in S1 Appendix. Missingness was low across all the covariates (1.9%). We therefore used complete-case analysis for the estimation of the beta coefficients. The magnitude of intra-pregnancy longitudinal changes was compared to the magnitude of biological variation (see Text F in S1 Appendix for details on biological variation) expected in healthy non-pregnant adults to highlight intra-pregnancy excursions from hematologic steady state outside of pregnancy [31].

## Associations with adverse outcomes

We assessed the associations between out-of-range CBC values at 26–29 weeks' gestation or rare longitudinal changes (between the 7–14 and 26–29 weeks' gestation timepoints) and pregnancy complications. We focused on the following complications that could plausibly be associated with hematologic abnormalities: preterm birth, small for gestational age birth weight (SGA), and hypertensive disorders of pregnancy (HDP) including pre-eclampsia. Preterm birth was defined as delivery between 29 and 36 weeks' gestation. HDP, including gestational hypertension and preeclampsia, were identified based on outpatient blood pressure measurements meeting the American Congress of Obstetricians and Gynecologists (ACOG) criteria [32] (two or more systolic readings ≥140 mmHg and/or diastolic readings ≥90 mmHg at least four hours apart after 20 weeks of gestation and up to one week postpartum), diagnosis documented in the delivery record, or at least one elevated blood pressure with laboratory evidence of pre-eclampsia [33] (see Text G in S1 Appendix). The HDP definition was validated via blinded chart review of a subset of medical charts in the discovery cohort with an accuracy of 90%. Because we evaluated our exposures at 26–29 weeks' gestation, we excluded pregnancies with HDP where the first high blood pressure occurred at <29 weeks' gestation. SGA was defined as <10th percentile for gestational age and sex using established thresholds (see Text H in S1 Appendix) [34]. In the discovery cohort, we also considered transfusion of plasma, platelets, and/or red blood cells at delivery or within 8 weeks postpartum. Our primary composite outcome included HDP, preterm birth, and SGA.

For association testing, pregnancies meeting inclusion criteria were divided into those with and without relevant complications. For each CBC index, a pregnancy would then be classified as out-of-range or having a rare dynamic (both described below). Associations quantified via odds ratios (ORs) were then computed with generalized estimating equations (GEE) for logistic regression in order to account for the lack of independence between pregnancies in the same individual [35–37]. Clustering was enforced at the level of the individual, and an exchangeable correlation structure was used. Models were adjusted for 1st trimester BMI, age, insurance status, race/ethnicity, and parity. Missingness was low across

covariates (3.7% in discovery and 10.7% in validation), and complete cases were used for association testing. Bonferroni correction of alpha = 0.05 was used to account for multiple hypothesis testing where appropriate, with the corresponding significance level specified in individual results (see Text I in S1 Appendix for additional detail).

Out-of-range CBC analysis was conducted in individuals who had a 26–29 week CBC in both the discovery and validation cohorts. The exposures, out-of-range CBC values, were defined as CBC values lying outside the 26–29 week reference interval derived in the discovery cohort. Out-of-range CBC values and adverse outcomes that were significantly associated in the discovery cohort were subsequently tested in the validation cohort.

Rare longitudinal changes analysis was conducted on the subset of individuals having both a 7–14 week and 26–29 week CBC in discovery and validation. Exposure was defined as a change in a CBC index between 7–14 weeks' and 26–29 weeks' gestation above or below a data-driven threshold identified in the discovery cohort. Direction (above or below) was chosen based on determination of the less common variation exceeding the established level of healthy non-pregnant biological variation, reflecting stable excursions from hematologic steady state in pregnancy in one direction (see Reference intervals and longitudinal dynamics in pregnancies without complications). One hundred equally spaced thresholds were obtained based on the distribution of changes from all available CBCs observed between 7–14 and 26–29 weeks' gestation in the discovery cohort. We chose the threshold that yielded the highest positive predictive value (PPV) and significant association for each outcome (more details available in Text J in S1 Appendix). We focused on PPV because it helps assess a marker's potential clinical utility, which rests upon PPV exceeding the prevalence of the outcome being predicted. All thresholds with PPVs whose confidence intervals did not include the prevalence of a given outcome in the discovery cohort potentially provided novel prognostic value and were tested in the validation cohort. This selection process is described in Fig 1.

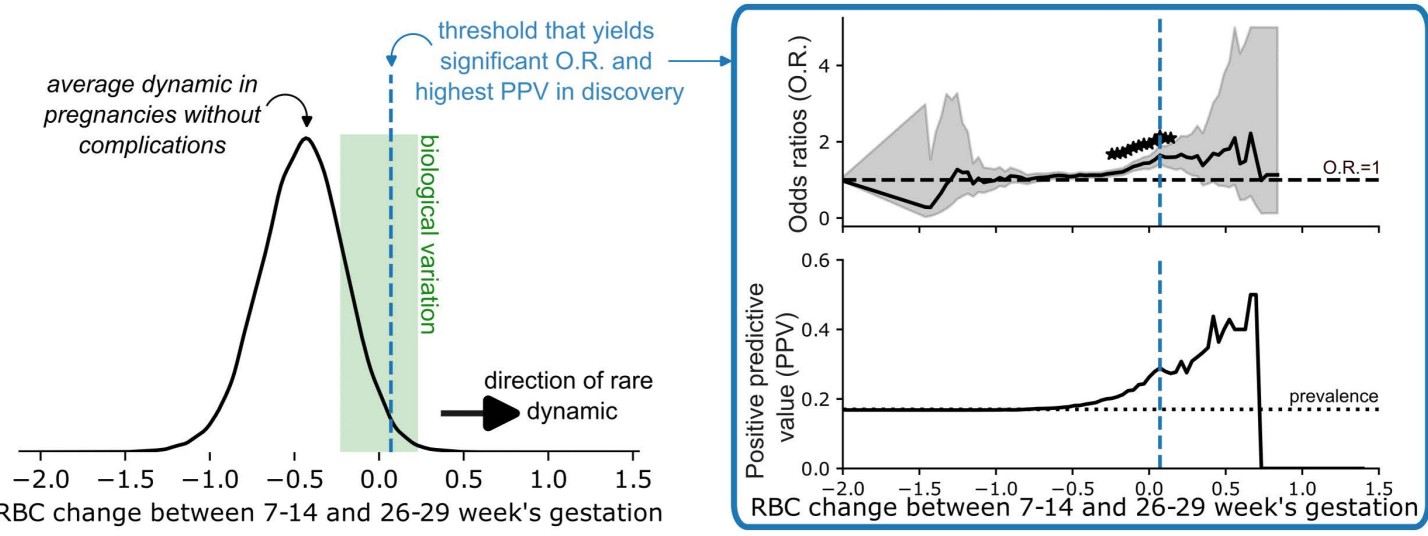

**Fig 1. Visual depiction of the definition of rare dynamics between 7–14 and 26–29 weeks' gestation.** In the example, we plotted the change for red blood cell count (RBC). On the left, we show the distribution of changes in term pregnancies without complications. The mean is negative and exceeds biological variation (green shaded area), implying that an increase in RBC is an unusual direction of change in pregnancy (right-pointing black arrow). To determine the threshold that we used to define our exposure, we looked at a series of thresholds across all observed changes and assigned the presence of the exposure to those values above the threshold based on the direction of rare dynamic. On the right side, we then evaluated the odds ratios (O.R.) and the positive predictive value (PPV) in discovery. We chose the threshold that led to a significant O.R. and the highest PPV (blue dashed line on both sides). This threshold for rare dynamics often falls within biological variation, implying that healthy physiological changes in pregnancy often exceed simple fluctuations of the markers. Abbreviations: O.R., odds ratios; PPV, positive predictive value, RBC, red blood cell count.

PLOS Medicine

To evaluate the robustness of the reported associations, we conducted three sensitivity analyses within the discovery cohort. First, we included pregnancies that had been excluded in the primary analysis due to stillbirth or underlying blood disorders. Stillbirths were added to the composite outcome, and the presence of a blood disorder was added to our GEE model as a covariate. Second, we additionally adjusted for covariates capturing potential immune- and inflammation-related confounding, including pre-existing or incident autoimmune conditions and infections during pregnancy (ascertained via Phecodes; see Text K in S1 Appendix), as well as past or current smoking status. Third, we restricted the analysis to pregnancies with no medication exposure or with exposure limited to medications not known to affect hematological dynamics (see Text L in S1 Appendix for the full medication list). For all three sensitivity analyses, associations between outcomes and either out-of-range CBC values or rare dynamics were tested and compared to associations from the primary analysis.

Analyses were performed in Python 3.11.3 and R version 4.3.1

## Ethics statement

The Mass General Brigham Institutional Review Board (MGB IRB) approved the study and waived the requirement for informed consent given that the research involved no more than minimal risk to participants. The study is conducted under MGB IRB protocols 2005P000660 and 2018P002283.

## Results

### Study population

The discovery cohort consisted of 45,992 pregnancies in 32,731 individuals (Fig 2 and Table 1). There were 37,709 (82%) term pregnancies without relevant complications and 8,283 (18%) pregnancies with the specified complications (preterm

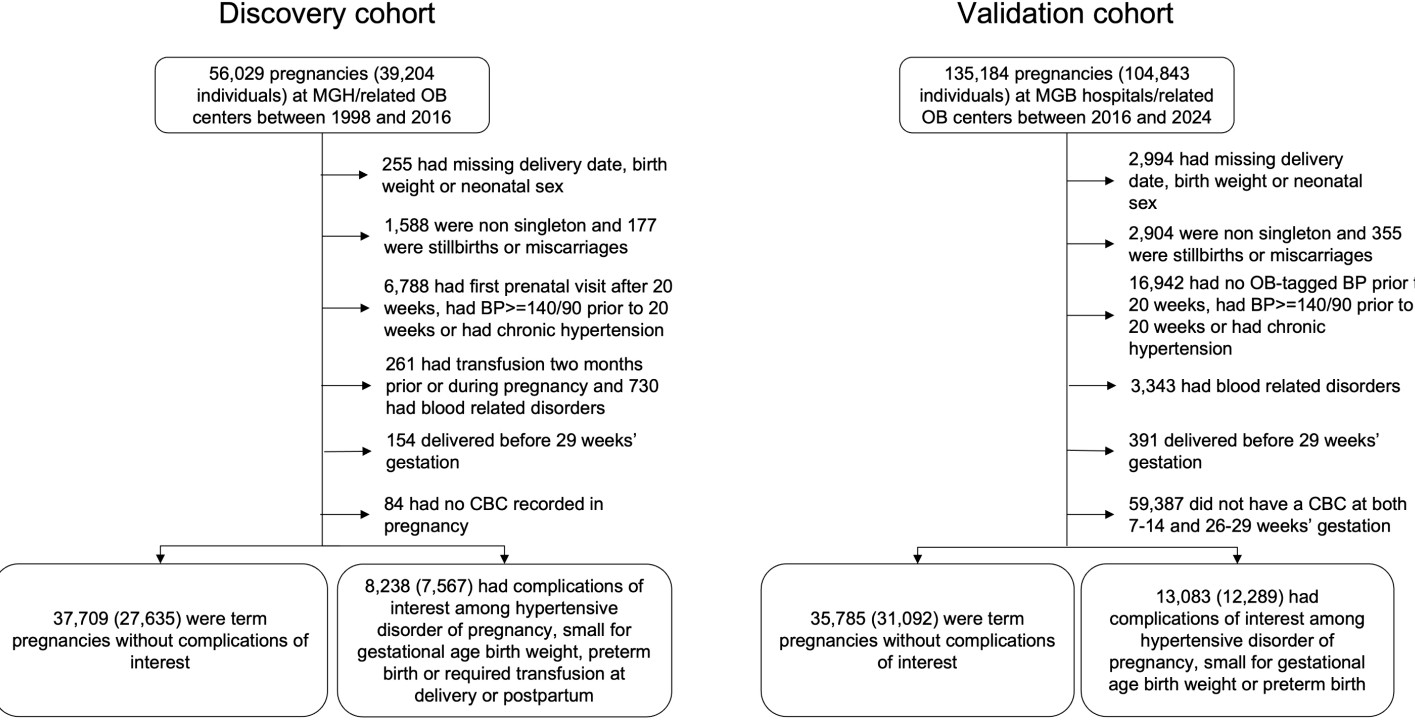

**Fig 2. Selection of pregnancies for the discovery cohort from the Maternal Health Cohort, and for the validation cohort from MGB-wide deliveries.** *Abbreviations:* MGH, Massachusetts General Hospital; MGB, Massachusetts General Brigham; CBC, Complete blood counts; OB, Obstetric.

**Table 1. Characteristics of the analyzed cohorts.**

| Variable | Discovery cohort (*N*=45,992) | | Validation cohort (*N*=48,868) | |
|---|---|---|---|---|
| | **Pregnancies without complications** | **Pregnancies with complications** | **Pregnancies without complications** | **Pregnancies with complications** |
| Pregnancies *N* (%) | 37,709 (100) | 8,283 (100) | 35,785 (100) | 13,083 (100) |
| Age, years, Mean (SD) | 32 (6) | 31 (6) | 34 (4) | 33 (5) |
| Insurance status, *N* (%) | | | | |
| *Public* | 9,400 (25) | 2,232 (27) | 4,309 (12) | 1,781 (14) |
| *Private* | 24,926 (66) | 5,396 (65) | 31,141 (87) | 11,194 (86) |
| *None/limited* | 3,383 (9) | 655 (8) | 335 (1) | 108 (1) |
| Self-reported race, *N* (%) | | | | |
| *Asian* | 3,256 (9) | 829 (10) | 4,131 (12) | 1,426 (11) |
| *Black* | 2,344 (6) | 657 (8) | 2,056 (6) | 996 (8) |
| *Latina* | 4,659 (12) | 962 (12) | 2,880 (8) | 1,217 (9) |
| *White* | 22,783 (60) | 4,844 (58) | 25,682 (72) | 9,069 (69) |
| *None of the above* | 4,667 (13) | 991 (12) | 1,036 (3) | 375 (3) |
| Nulliparous, N (%) | 16,709 (44) | 4,816 (58) | 15,331 (43) | 7,737 (59) |
| Gestational week at delivery, weeks, Mean (SD) | 40 (1) | 38 (2) | 40.0 (1) | 38.0 (2) |
| Prenatal BMI at 12 weeks, kg/m$^2$, mean (SD) | 25.2 (5.0) | 25.8 (5.6) | 25.7 (5.1) | 27.2 (6.2) |
| Time of CBC measurement | | | | |
| *Pre-pregnancy* | 3,980 (11) | 1,057 (13) | N/A | N/A |
| *7–14 weeks' gestation* | 29,126 (77) | 6,365 (77) | 35,785 (100) | 13,083 (100) |
| *26–29 weeks' gestation* | 27,941 (74) | 6,218 (75) | 35,785 (100) | 13,083 (100) |
| *Pre-delivery* | 37,528 (100) | 8,260 (100) | N/A | N/A |
| *Pre-pregnancy and 7–14 weeks' gestation* | 3,333 (9) | 868 (10) | N/A | N/A |
| *7–14 weeks and 26–29 weeks' gestation* | 24,125 (64) | 5,291 (64) | 35,785 (100) | 13,083 (100) |
| *26–29 weeks and Pre-delivery* | 27,795 (74) | 6,202 (74) | N/A | N/A |
| *All time points* | 2,791 (7) | 729 (9) | N/A | N/A |

BMI at 12 weeks' gestation was either measured, interpolated from available measurements, or extrapolated based on a population model. When no BMI measurement was available, it was treated as missing. See Supplementary Methods in S1 Appendix for more details.

CBC, Complete blood count.

Complications—hypertensive disorder of pregnancy, preeclampsia, small for gestational age, preterm delivery at <37 weeks' gestation, transfusion at or within 8 weeks after delivery.

birth, HDP, SGA, or transfusion at or after delivery, see S1 Table for detail). The validation cohort consisted of 48,868 pregnancies, of which 13,083 (27%) had a complication of interest (Fig 2 and Table 1); participants in the validation cohort had similar characteristics to those in the discovery cohort, but delivered in a different study period. We observed an increase in the proportion of complications of interest between validation and discovery (S1 Table), likely related to an increase in the prevalence of HDP that may reflect an epidemiological increase in HDP across our combined 26 year study period, and changes in HDP delivery note coding practices when the MGB hospital system adopted a new medical record system in 2016 [38]. Pregnancies with and without complications had similar baseline characteristics in both the discovery and validation cohorts, with the exception of nulliparity (Table 1). We analyzed the demographic characteristics of pregnancies that were excluded from the analysis (S2 Table), which show a higher number of pregnancies with public insurance, a higher number of individuals who self-identified as Black or Hispanic, and a higher BMI. Their exclusion could be due in part to socio-economic factors that led to reduced access to care and prevented us from collecting necessary CBC data for the analysis.

## Multiple CBC reference intervals differ from those in commonly cited literature

To derive new reference intervals, we analyzed distributions of CBC indices in pregnancies without complications in the discovery cohort (*N* = 37,709) and compared trimester-specific CBC index reference intervals to those in commonly cited literature and to those in regular use for non-pregnant adult females (Fig 3). All numerical values are provided in S3 Table. The reference intervals for HCT, HGB, WBC, PLT, and RDW were consistent with previously reported ranges. RBC, MCV, MCH, and MCHC reference intervals showed notable differences, with only 75%, 74%, 30%, and 52% of uncomplicated pregnancies in our cohort falling within literature reference intervals throughout gestation, meaning that according to current practices at least 25% and as many as 70% of uncomplicated pregnancies had at least one abnormal result for each of these CBC indices. This discrepancy was not due to the inclusion of pregnancies with anemia, pregnancies with prescribed oral or IV iron, nor those with chronic or pregnancy-related conditions (see Figs C, D, and E in S1 Appendix for more detail).

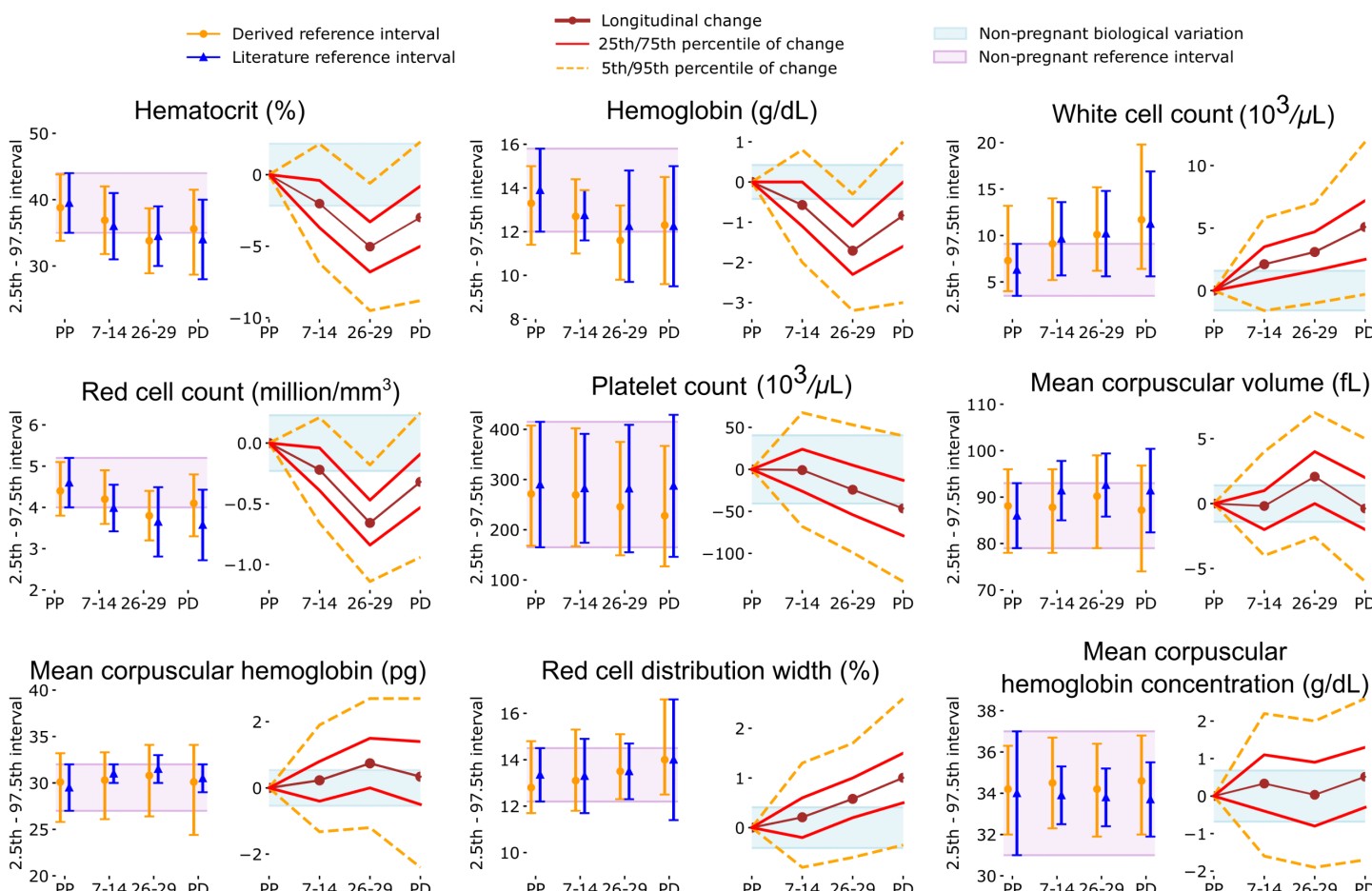

**Fig 3. Reference intervals and average values for CBC indices vary substantially across gestation, with high inter-individual variability.** For each marker, on the left are the gestational age-specific reference intervals from the Maternal Health Cohort (*N* = 37,709, orange error bars with round marker), in comparison with literature trimester-specific reference intervals (blue error bars with triangle). On the right are the intra-pregnancy longitudinal changes relative to pre-pregnancy baseline in *N* = 2,791 pregnancies for which CBCs were available at all considered time points. Literature and non-pregnant reference intervals, shaded areas, were retrieved from prior publications [28,29]. *Abbreviations:* PP is Pre-Pregnancy, and PD is Pre-Delivery at term. The numerical values can be found in S3, S4, and S5 Tables.

## CBC indices change dramatically during uncomplicated pregnancies

In our analysis of the subset of the uncomplicated term pregnancies that had all CBCs available at the 4 examined time-points (including prior to pregnancy and 3 pregnancy timepoints, $N=2,791$), CBC indices changed substantially over time (Fig 3; S4 and S5 Tables), with all but one intra-pregnancy CBC index change reaching statistical significance ($p<0.002$). The change between pre-pregnancy and 7–14 weeks' gestation for PLT was the only exception. For all CBC indices, 68%–98% of pregnancies changed during gestation by more than what is expected for healthy biological variation outside pregnancy in at least one time period (S6 Table). Most CBC indices decreased initially between pre-pregnancy and the 1st trimester. Then, between 7–14 weeks' and 26–29 weeks' gestation, HGB, RBC, and HCT decreased by an amount greater than biological variation in 82%, 81%, and 66% of pregnancies. WBC and PLT remained within biological variation in 54% and 66% of pregnancy, while MCV, MCH, RDW, and MCHC were typically either stable or increased between 7–14 weeks' and 26–29 weeks' gestation. From 26–29 weeks' gestation to pre-delivery (at term), HCT, HGB, WBC, RBC, RDW, and MCHC increased or were stable in ~90% of pregnancies, while PLT, MCV, and MCH decreased or were stable in 83–95% of pregnancies (S6 Table). Sensitivity analyses assessing typical longitudinal changes in pregnancies with CBCs in at least two consecutive timepoints were consistent with the primary analyses (the number of analyzed pregnancies is available in Table 1 and results in S7 Table).

## Elevated red cell indices are associated with complications

In discovery, we investigated the use of the 26–29 weeks' gestation reference intervals for the identification of pregnancies at elevated risk of subsequent adverse outcomes. Of those $N=34,159$ pregnancies with and without complications having a CBC at 26–29 weeks' gestation, pregnancies with HGB, HCT, or RBC above the reference interval at 26–29 weeks' gestation were significantly more likely to have the composite outcome (Fig 4 and S8 Table): OR 1.4 (95% CI [1.2, 1.7] $p<0.0001$) for HCT, 1.7 (95% CI [1.4, 1.9] $p<0.0001$) for HGB, and 1.6 (95% CI [1.4, 1.9] $p<0.0001$) for RBC. Elevated HGB and RBC at 26–29 weeks' gestation were associated with HDP with OR 1.7 (95% CI [1.4, 2.2], $p<0.0001$ for HGB and 1.7 (95% CI [1.4, 2.2] $p<0.0001$) for RBC; preeclampsia with OR 1.9 (95% CI [1.4, 2.5], $p<0.0001$) for HGB and 2.1 (95% CI [1.6, 2.8]), $p<0.0001$) for RBC, and preterm birth with OR: 1.8 (95% CI [1.4, 2.3], $p<0.0001$) for HGB and 2.0 (95% CI [1.6, 2.5] $p<0.0001$) for RBC when these outcomes were examined individually (Fig 4 and S8 Table). PLT was not significantly associated with any outcomes, but values above the reference interval were nominally significantly associated with the composite outcome and HDP. CBC values below the 26–29 weeks' gestation reference interval were not significantly associated with the composite outcome or its individual components, though RBC values below the reference interval were nominally significantly associated with HDP. There were no significant associations with SGA, and peripartum transfusion was significantly associated with HGB below the reference interval (OR: 3.4 [2.0, 5.8] $p<0.0001$, S8 Table). Among the covariates, maternal age, BMI, and public insurance status were at least nominally significantly associated with an increased risk of the composite outcome, with modest effect sizes. In contrast, parity showed a strong and consistently protective association. No significant associations were observed for race and ethnicity. Effects were consistent across CBC parameters and exposures (S9 Table).

Results in the validation cohort ($N=48,868$) were consistent with those in the discovery cohort (Fig 4 and S10 Table). In particular, pregnancies with HGB, HCT, or RBC above the reference interval at 26–29 weeks' gestation were significantly more likely to have the composite outcome: OR 1.4 (95% CI [1.2, 1.5] $p<0.0001$) for HCT, 1.6 (95% CI [1.4, 1.8] $p<0.0001$) for HGB, and 1.6 (95% CI [1.4, 1.7] $p<0.0001$) for RBC. Elevated HGB, HCT, and RBC at 26–29 weeks' gestation were associated with HDP with ORs for HCT of 1.5 (95% CI [1.3, 1.6], $p<0.0001$); for HGB 1.7 (95% CI [1.4, 2.0], $p<0.0001$); and for RBC 1.5 (95% CI [1.3, 1.7] $p<0.0001$). RBC above range was also consistently significantly associated with preterm birth: OR1.6 (95% CI [1.3, 1.9], $p<0.0001$). RBC below range was significantly associated with a reduction in HDP: OR 0.7 (95% CI [0.6, 0.8], $p<0.0001$).

Sensitivity analysis in the discovery cohort, including pregnancies resulting in stillbirth or reporting blood disorders (S11 Table), adding autoimmune conditions, infections, and smoking status (S12 Table), or restricting to those

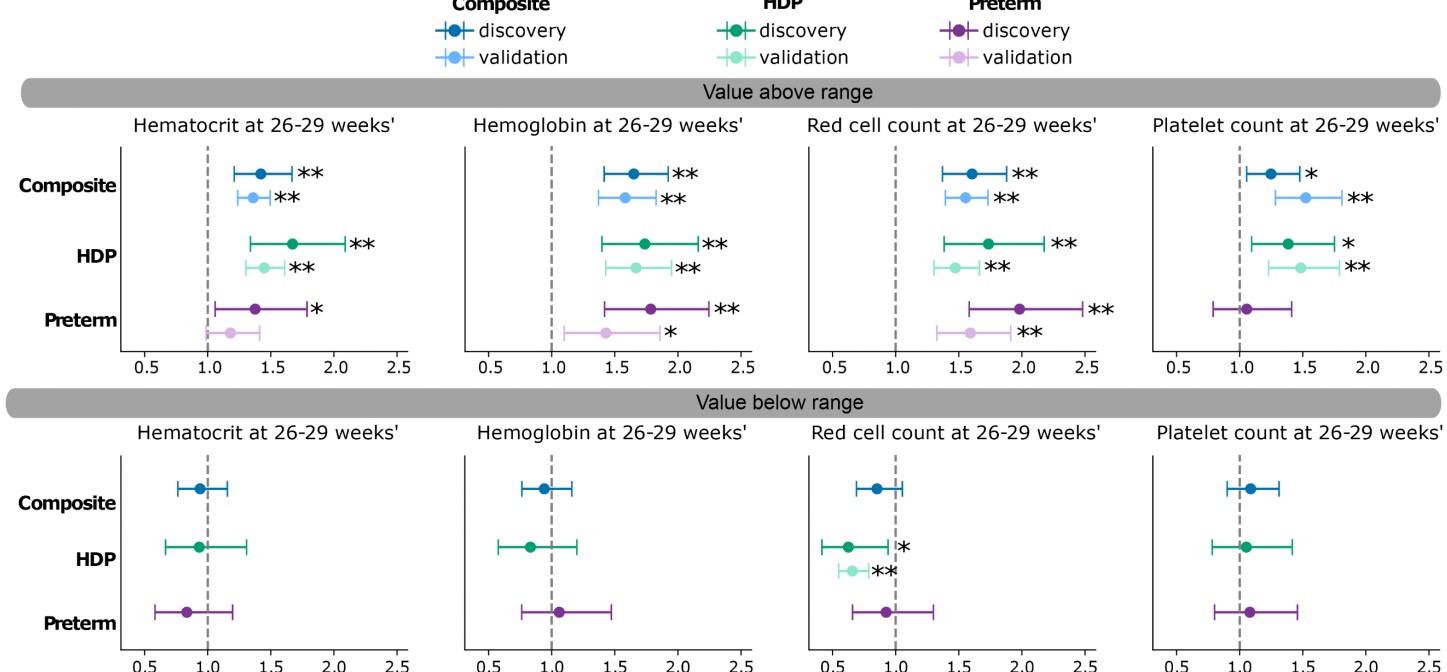

**Fig 4. CBC values above their reference intervals are associated with complications.** Odds ratios for developing the composite outcome (hypertensive disorders of pregnancy [HDP], small for gestational age birth weight [SGA], preterm birth) and individual complications are shown. Pregnancies with a CBC at 26–29 weeks' gestation were considered ($N = 34,159$ in discovery, $N = 48,868$ in validation), and those with preterm birth at <30 weeks' gestation or evidence of HDP or preeclampsia before 29 weeks were excluded ($N = 297$ in discovery, $N = 378$ in validation). Odds ratios are marked as significant in discovery with ** if the Wald test $p$-value is below 0.0003 in discovery and below 0.004 in validation, reflecting a Bonferroni correction. Odds ratios are marked with * if they are nominally significant with a $p$-value below 0.05. The numerical values can be found in S8 and S10 Tables.

pregnancies that did not report medications possibly affecting CBC dynamics (S13 Table), showed consistent results with the primary analysis (Fig F in S1 Appendix).

### Rare CBC index changes are associated with complications

Given the large variation in CBC indices during pregnancy, we tested whether rare changes in CBC indices were associated with the risk of subsequent complications in participants who had CBCs at 7–14 and 26–29 weeks' gestation ($N = 29,416$ discovery, $N = 48,868$ validation). We defined rare longitudinal changes for each CBC index based on the distribution of these changes in the discovery cohort (Fig 1). Between 7–14 and 26–29 weeks' gestation, the least frequent changes exceeding biological variation in magnitude were an increase in HGB (2% of pregnancies), HCT (1.3%), RBC (0.4%), PLT (4%), or MCHC (18%), and a decrease in WBC (8.4%), MCV (5.7%), MCH (13.4%), or RDW (10.1%) (Fig 3 and S6 Table). In the discovery cohort, rare changes were associated with increased risk of subsequent complications. Between 7–14 and 26–29 weeks' gestation, an increase of >1.8% in HCT was associated with the composite outcome with an OR of 1.6 (95% CI [1.3, 1.9], p<0,0001); as was an increase of >0.67 g/dL in HGB with OR2.0 (95% CI [1.6, 2.5], p<0.0001), an increase >0.07 $10^6$/mm$^3$ in RBC with OR1.9 (95% CI [1.6, 2.4], p<0.0001), a decrease of less than 0.05 $10^3$/µl or an increase in PLT with OR1.2 (95% CI [1.1, 1.3]), $p < 0.0001$), and a decrease in MCV greater than 0.75 fL with OR 1.3 (95% CI [1.2, 1.4], $p < 0.0001$, see Tables 2 and S14). No significant associations were found between rare longitudinal changes and peripartum transfusion (S14 Table).

Consistent results were also found in the validation cohort using the thresholds defined in the discovery cohort (Tables 2 and S15). For the composite outcome, an increase of >1.8% in HCT was significantly associated with an OR

**Table 2. Significant associations between rare longitudinal CBC changes and obstetric complications.**

| Complication Prevalence [95% CI] | CBC Index | Threshold | Validation Cohort | | | Discovery Cohort |
|---|---|---|---|---|---|---|
| | | | OR [95% CI] | PPV [95% CI] | % identified in range [95% CI] | OR [95% CI] |
| Composite 26.2% [25.8, 26.6] | Hematocrit | ↑ greater than 1.8% | 1.46 [1.28, 1.67] | 34.0 [31.1, 36.8] | 71 [66, 76] | 1.59 [1.31, 1.94] |
| | Hemoglobin | ↑ greater than 0.67 g/dL | 1.52 [1.28, 1.8] | 35.6 [32.1, 39.4] | 83 [78, 88] | 1.99 [1.57, 2.52] |
| | Red cell count | ↑ greater than 0.07 $10^6$/mm³ | 1.72 [1.5, 1.97] | 38.4 [35.4, 41.6] | 79 [75, 84] | 1.94 [1.6, 2.35] |
| | Platelet count | ↓ of less than 0.05 $10^3$/μL or ↑ | 1.09 [1.04, 1.15] | 28.3 [27.4, 29.2] | 94 [93, 95] | 1.20 [1.12, 1.29] |
| | Mean corpuscular volume | ↓ greater than 0.75fL | 1.28 [1.19, 1.39] | 32.6 [31.1, 34.1] | 90 [88, 92] | 1.32 [1.2, 1.45] |
| HDP 16.9% [16.5, 17.2] | Hematocrit | ↓ of less than 0.05% or ↑ | 1.17 [1.08, 1.27] | 18.6 [17.6, 19.7] | 80 [77, 82] | 1.40 [1.21, 1.62] |
| | Hemoglobin | ↑ greater than 0.11 g/dL | 1.27 [1.14, 1.42] | 20.1 [18.5, 21.8] | 85 [82, 89] | 1.66 [1.36, 2.02] |
| | Red cell count | ↓ of less than 0.034 $10^6$/mm³ or ↑ | 1.42 [1.27, 1.57] | 22.5 [20.7, 24.1] | 83 [80, 86] | 1.54 [1.28, 1.85] |
| | Mean corpuscular volume | ↓ greater than 0.75fL | 1.23 [1.12, 1.34] | 22.0 [20.7, 23.3] | 92 [90, 93] | 1.35 [1.18, 1.55] |
| Preterm 5.5% [5.3, 5.7] | Hematocrit | ↑ greater than 0.26% | 1.39 [1.22, 1.59] | 7.6 [6.8, 8.4] | 81 [76, 85] | 1.47 [1.24, 1.74] |
| | Hemoglobin | ↑ greater than 0.67 g/dL | 1.92 [1.47, 2.51] | 10.7 [8.3, 13.2] | 78 [68, 88] | 2.28 [1.62, 3.2] |
| | Red cell count | ↑ greater than 0.071 $10^6$/mm³ | 2.07 [1.68, 2.57] | 11.1 [9.2, 13.1] | 77 [68, 84] | 2.33 [1.77, 3.07] |
| | Mean corpuscular volume | ↓ greater than 0.75fL | 1.53 [1.34, 1.74] | 8.4 [7.5, 9.4] | 89 [85, 93] | 1.41 [1.22, 1.63] |
| SGA 7.3% [7.1, 7.6] | Hematocrit | ↑ greater than 1.19% | 1.59 [1.38, 1.85] | 11.7 [10.4, 13.1] | 78 [72, 83] | 1.67 [1.35, 2.05] |
| | Hemoglobin | ↑ greater than 0.22 g/dL | 1.60 [1.37, 1.86] | 11.9 [10.3, 13.4] | 87 [83, 92] | 1.64 [1.34, 2.01] |
| | Red cell count | ↑ greater than 0.071 $10^6$/mm³ | 1.86 [1.54, 2.26] | 13.4 [11.3, 15.4] | 82 [76, 89] | 1.92 [1.47, 2.51] |

Pregnancies with rare longitudinal changes in some CBC indices between 7–14 and 26–29 weeks' gestation have higher rates of complications. Those with preterm birth at <30 weeks' gestation or evidence of HDP or preeclampsia before 29 weeks were excluded ($N = 297$ in discovery, $N = 378$ in validation from original $N = 29{,}416$ and $N = 48{,}868$). Listed are the OR and PPV in our validation cohort of the rare changes significantly associated with adverse outcomes ($p < 0.003$ for validation, $p < 9*10^{-6}$ for discovery from a Wald test, reflecting Bonferroni correction) and for which the positive predictive value (PPV) confidence interval did not overlap prevalence confidence intervals in our discovery cohort. Confidence intervals were obtained with bootstrapping. PPV, positive predictive value; % identified in range, percentage of pregnancies that fall within the derived reference intervals at 26–29 weeks' gestation with adverse outcomes correctly identified by a rare longitudinal change; SGA, small for gestational age; HDP, hypertensive disorder of pregnancy.

of 1.5 (95% CI [1.3, 1.7] $p < 0.0001$), as was an increase of >0.67 g/dL in HGB, with an OR of 1.5 (95% CI [1.3, 1.8] $p < 0.0001$), an increase >0.07 $10^6$/mm³ in RBC had an OR of 1.7 (95% CI [1.5, 2.0] $p < 0.0001$), a decrease of less than 0.05 $10^3$/μl or an increase in PLT had an OR of 1.1 (95% CI [1.0, 1.2], $p = 0.0008$), and a decrease in MCV greater than 0.75 fL had an OR of 1.3 (95% CI [1.2, 1.5], $p < 0.0001$). For the majority of pregnancies with complications identified by

these rare changes, the relevant CBC index value was within the 26–29 weeks' gestation reference interval (71%–93%) and was therefore unremarkable on its own (Table 2).

We analyzed each complication separately and found significant associations between preterm birth and rare changes in HGB or RBC in the discovery cohort; both associations were subsequently confirmed in the validation cohort. The preterm birth OR in the validation cohort was 1.9 for HGB (95% CI [1.5, 2.5] $p<0.0001$) and 2.1 for RBC (95% CI [1.7, 2.6] $p<0.0001$) for threshold increases of ≥0.67 (g/dL) in HGB or ≥0.07 ($10^6$/mm$^3$) in RBC (Table 2). PPVs for these rare changes were about twice the prevalence of preterm birth in the population. The subsequent preterm deliveries associated with these rare CBC changes included both spontaneous and medically indicated deliveries, and the proportion of each type of preterm delivery predicted was not distinguishable from the proportion in the full cohort. Most preterm deliveries (76%–80%) associated with these rare changes had HGB and RBC within the 26–29 weeks' gestation reference intervals and would not typically be identified based on that reference interval alone (Table 2). To explore the potential for early risk stratification, we analyzed the time intervals between detection of the rare behavior and preterm birth and found that, on average, it was >7 weeks for both the discovery cohort and the validation cohort (7.9 weeks for HGB and 7.5 for RBC).

We evaluated the robustness of the thresholds identified in the discovery cohort through three sensitivity analyses. These included: (1) expanding the cohort to include pregnancies complicated by stillbirth or blood disorders (S16 Table); (2) additionally adjusting for autoimmune conditions, infections, and smoking status (S17 Table); and (3) restricting the analysis to pregnancies not exposed to medications that could plausibly affect CBC dynamics (S18 Table). Across all analyses, results were consistent with the primary findings (Fig F in S1 Appendix). Because iron supplementation initiated after a diagnosis of anemia at 7–14 weeks' gestation could potentially influence CBC values, we also assessed the PPV of the identified thresholds separately in pregnancies with and without anemia. PPVs were consistent across these groups, indicating that the predictive performance of the thresholds was not driven by early anemia or its treatment (S19 Table).

## Discussion

Here, we derive gestational age-specific reference intervals for CBC indices and their changes during pregnancy in a large retrospective study of more than 94,000 pregnancies. Using these reference intervals, we found that pregnancies in our study cohort with elevations (either absolute or relative to baseline) in certain CBC indices were at significantly greater risk of obstetric complications. In particular, elevations in red blood cell-related indices were associated with increased risk of subsequent HDP and preterm birth. We also found a doubling in risk of preterm birth for pregnancies with uncommon increases in HGB or RBC between 7–14 and 26–29 weeks' gestation. These findings highlight an opportunity to improve risk stratification for pregnant individuals by systematically assessing longitudinal changes in routinely available CBC indices.

The reference intervals identified in our cohort were noticeably wider than previously published trimester-specific intervals for RBC, MCV, MCH, and MCHC [3,4,12,18,19,39]. These differences may in part reflect the smaller size of previous study cohorts [3,4,28,]. Recent larger studies outside the US found wider intervals for MCV and MCH, consistent with our results [5,40,41]. It is noteworthy that this study cohort's HGB reference intervals include values low enough to be considered anemic according to some current guidelines [30]. This difference may arise because current HGB thresholds for clinical management of anemia were, in some cases, derived from studies in which participants were iron-supplemented, potentially affecting their applicability to all pregnant individuals [42]. The absence of association in our study between decreased red cell indices and adverse outcomes may be surprising and may have been affected by the fact that individuals with low HGB or HCT values in our study were likely supplemented with iron based on current recommendations [43–45]. However, associations between increasing red cell indices and adverse complications are not driven by early anemia or iron supplementation, as indicated by our sensitivity analysis on PPVs of pregnancies with and without anemia.

Overall, we found that red cell values above the reference interval were significantly associated with adverse outcomes compared to values below. The findings that elevations in HCT, HGB, and RBC were associated with a higher risk of

preeclampsia, HDP, SGA, and preterm birth are consistent with a smaller study, which found that higher first-trimester HGB concentrations were associated with an increased risk of gestational hypertension, preeclampsia, preterm birth, and SGA [46]. Three other smaller studies have linked higher HGB concentrations to stillbirths, HDP, and lower birth weight [8,47,48]. We speculate that high HGB values might be a sign of hemoconcentration or might be associated with abnormal placentation, both of which have been hypothesized to be involved in the pathophysiology of HDP, SGA, and preterm birth [46,49]. We also found that, although not significant after multiple testing, elevated platelet (PLT) count was nominally significantly associated with the composite outcomes and HDP, in line with previous findings on platelets dynamics during pregnancy [17]. Current guidelines on CBC interpretation during pregnancy focus primarily on low values, which in our analysis were not significantly associated with adverse outcomes. Previous studies suggest that the timing of assessment may influence observed associations [17,50]. Our results, derived from a narrower gestational window than most previous studies, indicate that it might be beneficial to consider high CBC values at this specific stage of pregnancy as early indications of subsequent complications. Because CBCs are already routinely collected, incorporating the evaluation of high values may offer a cost-effective approach to detecting high-risk pregnancies.

CBC changes during pregnancy were often substantial, for instance with HGB and RBC for most pregnancies decreasing from baseline by more than twice what is expected for normal biological variation outside pregnancy [31]. These changes are consistent with those reported in previous studies, which were smaller or cross-sectional [3,15,17–19,21–23]. By assessing the magnitude and direction of a change in a CBC index, each patient is used as their own baseline, an approach which has been shown to be informative outside pregnancy [7]. The intra-pregnancy CBC changes reported in this study can help identify deviations that are not likely to be detected if the absolute CBC results remain within reference. Indeed, most pregnancies with rare changes and subsequent complications had CBC indices within the 26–29 weeks' reference interval.

We found that the rare increases in RBC or HGB between 7–14 and 26–29 weeks' gestation were associated with increased risk of the examined pregnancy complications, including preterm birth. The majority of pregnancies without complications had a decrease in RBC and HGB during this gestational period, presumably reflecting modulations of red cell mass and plasma volume typical of healthy pregnancies. The association of increases in red cell dynamics with complications may therefore be related to hemoconcentration, which has been suggested to be associated with obstetric complications in previous studies [46,49,51]. We also found an association between an uncommon decrease in MCV and the composite outcome, and this decreased MCV may reflect sub-clinical iron deficiency that may have been undetected and untreated, but further studies assessing iron status are necessary.

Our study has several strengths. First, we defined gestational-age-specific intervals in a cohort in the US that is significantly larger than those typically used in prior studies that did not involve meta-analysis [3,4,28]. Second, we systematically analyzed longitudinal intra-pregnancy CBC dynamics and explored their use as prognostic factors for pregnancy complications. Third, our validated definitions of pregnancy complications were derived using rigorous methods and are likely more reliable than those based solely on administrative data [24]. Fourth, our key findings were validated in an out-of-sample cohort, reducing the possibility of type I error.

Our study also has limitations. First, it was retrospective, and our discovery cohort was limited to pregnancies with prenatal care starting no later than 20 weeks' of gestation, and included some pregnancies with anemia or other pathologies. While our sensitivity analyses did not show major changes when anemic individuals were excluded, further investigations are warranted. Second, our findings regarding CBC-associated risk considered only uncomplicated pregnancies or those that developed complications after 29 weeks' gestation, and therefore these results may not generalize beyond this target population to pregnancies with complications occurring prior to 29 weeks. Third, we evaluated typical longitudinal changes in pregnancy at three time points, but more frequent time sampling under controlled conditions could be more accurate. In particular, because the discovery cohort was comprised of individuals who received obstetric care in our health system, not all participants had pre-pregnancy CBC data, which limited our sample size for examining typical longitudinal changes

in CBC indices that occur between pre-pregnancy and the first trimester. Thus, our analysis was limited out-of-range and rare behavior of CBCs at prenatal visits with higher CBC coverage (7–14 and 26–29 weeks' gestation). Fourth, discovery and validation cohorts spanned different time periods with evolving practices and diagnostic coding, leading to differences in prevalences and possible misalignment [38]. However, the consistent replication of associations across cohorts suggests that the identified biological signal is robust. Fifth, we evaluated univariate associations between CBC and obstetric complications, and it is possible that a multivariate approach could lead to stronger findings. Finally, the study population and chosen time windows might not accurately reflect obstetric practices outside of our health system.

In this study, we derive gestational-age-specific reference intervals for CBC indices and establish typical longitudinal changes in CBC indices across gestation in a large US-based cohort. We find that elevations and rare longitudinal increases in red cell indices are associated with complications of pregnancy. Future work is needed to determine whether these findings can help shed light on the physiologic processes that contribute to obstetric complications. Future prospective studies should determine whether assessment of longitudinal changes in CBC indices can be integrated into clinical practice to improve risk stratification.

## Supporting information

**S1 Checklist. RECORD guideline checklist.**
(DOCX)

**S1 Appendix. Supplementary Methods. Text A.** List of hematology-related problems used for exclusion. **Text B.** Detail on instruments used to measure Complete Blood Counts. **Text C.** Detail on inclusion criteria for routine pre-pregnancy CBCs. **Text D.** Estimation of body mass index (BMI) in discovery and validation cohorts. **Text E.** Mathematical formulation of mixed-effect model for longitudinal dynamics. **Text F.** Calculation of biological variation. **Text G.** Definition of hypertensive disorders of pregnancy. **Text H.** Definition of small for gestational age. **Text I.** Mathematical formulation for association analysis with adverse outcomes. **Text J**. Rare dynamic definition. **Text K.** List of phecodes used to detect pre-existing and novel autoimmune conditions and perinatal infections. **Text L.** List of medications not likely to affect complete blood counts used to define subset for sensitivity analysis. **Fig A. 95% intervals of complete blood count (CBC) indices of interest on the Siemens Advia 2120, Sysmex XE-5000 instruments, and Sysmex XN-9000 instruments.** Discrepancies between machines were so great for MPV that it was subsequently excluded from all further analysis. *Abbreviations:* HCT,hematocrit; HGB, hemoglobin; WBC, white cell count; RBC, red cell count; PLT, platelet count; MCV, mean corpuscular volume; MCH, mean corpuscular hemoglobin; MCHC, mean corpuscular hemoglobin concentration; MPV, mean platelet volume. **Fig B. Frequency of available CBCs by gestational week in discovery cohort pregnancies.** Shaded red areas indicate chosen windows for which we considered CBCs in gestation (7–14 weeks and 26–29 weeks). The pre-delivery time point is not shaded as it is individualized and fell between 30 and 41 weeks gestation. **Fig C. Sensitivity analysis of the effects of including patients with a diagnosis of anemia on reference interval determination.** The figure compares gestational-age-specific intervals for all term pregnancies without complications (blue error bars), pregnancies with no diagnosis of anemia (<11 g/dL HGB at PP, 7–14 and PD, and <10.5 g/dL at 26–29 weeks, orange error bars), pregnancies considered anemic (green error bars), and literature trimester-specific intervals (red error bars), all in the discovery cohort. Up to 15% of pregnancies met criteria for anemia at each time point. *Abbreviations:* HCT, hematocrit, HGB, hemoglobin; WBC, white cell count; RBC, red cell count; PLT, platelet count; MCV, mean corpuscular volume; MCH, mean corpuscular hemoglobin; MCHC, mean corpuscular hemoglobin concentration; MPV, mean platelet volume. **Fig D. Sensitivity analysis of the effects of including patients with recorded iron supplementation on reference interval determination.** The figure compares gestational-age-specific intervals for all term pregnancies without complications (blue error bars), pregnancies with no record of either oral or IV iron supplementation in their medical record (orange bars), and pregnancies with recorded oral or IV iron supplementation (green error bars, all in the discovery cohort.

*N* = 1,503 of 37,709 term pregnancies without complications had recorded IV or oral iron in their medical record. *Abbreviations:* HCT, hematocrit; HGB, hemoglobin; WBC, white cell count; RBC, red cell count; PLT, platelet count; MCV, mean corpuscular volume; MCH, mean corpuscular hemoglobin; MCHC, mean corpuscular hemoglobin concentration; MPV, mean platelet volume. **Fig E. Sensitivity analysis of the effects of including patients with chronic or pregnancy-related conditions on reference intervals.** Comparison of gestational-age-specific intervals for term pregnancies without complications in discovery, pregnancies in individuals without chronic or pregnancy-related conditions (*N* = 1,460) in discovery and literature trimester-specific intervals. *Abbreviations:* HCT, hematocrit; HGB, hemoglobin; WBC, white cell count; RBC, red cell count; PLT, platelet count; MCV, mean corpuscular volume; MCH, mean corpuscular hemoglobin; MCHC, mean corpuscular hemoglobin concentration; MPV, mean platelet volume. **Fig F. Odds ratio comparison between main analysis and sensitivity analyses for out of range associations (panel A) and rare dynamics (panel B).** Circled in black those associations that were significant in both main and sensitivity analysis. Depicted odds ratios are limited to those previously found significant in the main analysis. The hemoglobin (HGB) outlier can be attributed to a larger confidence interval [1.49, 26.49] that no longer meets Bonferroni correction, but is still nominally significant. *Abbreviations:* HCT, hematocrit, HGB, hemoglobin, WBC, white cell count; RBC, red cell count; PLT, platelet count; MCV, mean corpuscular volume; MCH, mean corpuscular hemoglobin; MCHC, mean corpuscular hemoglobin concentration; MPV, mean platelet volume; HDP, hypertensive disorder of pregnancy; SGA, small for gestational age; O.R., Odds ratio. **Fig G. Histogram of 50 most frequent PheCodes for visits corresponding to the pre-pregnancy CBCs considered.** Considered pre-pregnancy CBCs are described in Text C.
(DOCX)

**S1 Table. The number of total pregnancies and the number of pregnancies with various complications in both the discovery and validation cohorts.** Pregnancies with a first elevated blood pressure before 29 weeks' gestation are excluded from the group with a CBC at 26–29 weeks' gestation and the group with CBCs at both 7–14 and 26–29 weeks' gestation. *Abbreviations:* CBC, complete blood count; HDP, hypertensive disorders of pregnancy.
(XLSX)

**S2 Table. Comparison of demographics included/excluded pregnancies during the development of discovery and validation cohorts.** *Abbreviations:* BMI, body mass index.
(XLSX)

**S3 Table. Tabular version of reference intervals derived from discovery cohort and literature intervals.**
(XLSX)

**S4 Table. Beta coefficients of CBC dynamics in pregnancy.** Beta coefficients measure the effect of gestational age on CBC values and were estimated on the subset of pregnancies with all four time points available (*N* = 2,791). Random effects were added for the individual pregnancy, the individual, parity, OB-GYN site of care, and for year of delivery. *P*-values are calculated from the *t*-distribution using *t*-statistics generated from calculated beta cofficients divided by the calculated standard error for each coefficient. The *p*-value threshold with Bonferroni correction for multiple testing was *p* = 0.002.
(XLSX)

**S5 Table. Percentile changes of CBC indices in pregnancy.** PP is pre-pregnancy, PD is pre-delivery.
(XLSX)

**S6 Table. Rare CBC changes (<5%) during pregnancy occur for red blood cell count, hematocrit, hemoglobin, platelet count, and mean corpuscular volume.** Percentages of pregnancies with decreasing, stable, or increasing CBC indices between subsequent timepoints for analysis of common and uncommon behavior in blood dynamics

during pregnancy are shown. We considered a difference between two timepoints to be stable if the absolute value of the change was less than the biological variation for that index, increasing if it was positive and greater than biological variation, and decreasing if it was negative and greater in magnitude than biological variation (see S1 Appendix Text F for further details). Additionally, percentage of pregnancies with stable (within biological variation) indices throughout gestation are reported in the last column. The CBC indices for which fewer than 5% of pregnancies remain stable are bolded.
(XLSX)

**S7 Table. Beta coefficients are consistent when inferred on all available pregnancies with CBCs for at least two consecutive time points.** Beta coefficients were inferred on the subset of pregnancies without complications for which at least one consecutive pair of time points was available ($N=28,450$) and compared to the beta coefficients in the main analysis. *P*-values are calculated from the t-distribution using t-statistics generated from calculated beta cofficients divided by the calculated standard error for each coefficient. *P*-values were corrected for multiple testing with Bonferroni correction so that $p<0.002$ was considered significant. All absolute differences in beta coefficients between sensitivity and main analysis fall within 2× biological variation.
(XLSX)

**S8 Table. Elevated CBC index values are associated with adverse pregnancy outcomes.** Pregnancies with a CBC at 26–29 weeks' gestation were considered ($N=34,159$), and those with abnormal tests for HDP or preeclampsia diagnosis before 29 weeks' gestation were excluded ($N=254$). The odds ratios of developing examined complications (composite of HDP, SGA, and preterm delivery; each of those adverse events individually; and need for transfusion at or after delivery) are shown in table for a CBC index value above, outside, or below the study intervals at 26–29 weeks' gestation. Significant odds ratios are marked with a 1 in the corresponding column. Prevalence and positive predictive value (PPV) are also reported. *P*-values are generated from Wald tests. Significance was evaluated with a Bonferroni-corrected *p*-value of 0.0003. *Abbreviations:* PPV, positive predictive value; HDP, hypertensive disorder of pregnancy; SGA, small for gestational age.
(XLSX)

**S9 Table. Parity, and to a lesser extent maternal age, BMI, and public insurance status, were significantly associated with predicting the composite outcome in out-of-range analysis of discovery cohort.** Pregnancies with a CBC at 26–29 weeks' gestation were considered ($N=34,159$), and those with abnormal tests for HDP or preeclampsia diagnosis before 29 weeks' gestation were excluded ($N=254$). OR and p-values for all covariates used in our logisitic regression for each CBC index are reported in the table. *P*-values are generated from Wald tests. Abbreviations: BMI, Body Mass Index; OR, Odds ratio.
(XLSX)

**S10 Table. Odds ratios and PPVs for extreme CBC index values are consistent in an out-of-sample validation cohort.** Pregnancies with a CBC at 26–29 weeks' gestation were considered ($N=48,490$), and those with abnormal tests for HDP or preeclampsia diagnosis before 29 weeks' gestation were excluded ($N=387$). We analyzed associations between adverse outcomes and the presence of a CBC index above, outside, or below the study reference intervals derived in discovery at 26–29 weeks' gestation. Only combinations of CBC index/adverse outcomes that were at least nominally significant in the discovery cohort were considered for this analysis. In the table, we report ORs, prevalence, positive predictive value (PPV), and negative predictive value (NPV). *P*-values are generated from Wald tests. Significance was evaluated with a Bonferroni corrected *p*-value of 0.004 and significant ORs in validation are marked with a 1 in the corresponding column.
(XLSX)

**S11 Table. Sensitivity analysis of associations on an expanded discovery cohort including pregnancies ending in stillbirth and blood disorders previously excluded yielded near identical OR in out-of-range analysis.** In this sensivitiy analysis, the discovery cohort was re-derived to include pregnancies ending in stillbirth and blood disorders ($N = 41{,}153$). Pregnancies ending in stillbirth were labelled as having the composite outcome, and marked as preterm or HDP when applicable. The presence of a blood-related disorder was added as a binary covariate to our logistic regression. Odds ratios, positive predictive values and negative predictive values for each CBC index, outcome pair are listed in the table, as is the prevalence of each outcome. *P*-values are generated from Wald tests. Significance is denoted if $p < 0.0003$. *Abbreviations:* PPV, positive predictive value; NPV, negative predictive value; HDP, hypertensive disorder of pregnancy; SGA, small for gestational age.
(XLSX)

**S12 Table. Sensitivity analysis of associations in the discovery cohort ($N = 34{,}159$) including the presence/absence of smoking, auto-immune disorders, and infections during pregnancy yielded near identical OR in out-of-range analysis.** In this sensitivity analysis, information on infections during pregnancy, pre-existing auto-immune disorders and smoking status was acquired from the electronic health record, and binary covariates were developed reflecting their presence or absence (see S1 Appendix Text K for details). Odds ratios, positive predictive values, and negative predictive values for each CBC index, outcome pair are listed in the table, as is the prevalence of each outcome. *P*-values are generated from Wald tests. Significance is denoted if $p < 0.0003$. *Abbreviations:* PPV, Positive Predictive value; HDP, Hypertensive disorder of pregnancy; SGA, Small for gestational age.
(XLSX)

**S13 Table. Sensitivity analysis of associations in a subset of the discovery cohort ($N = 6{,}709$) with no medications that may cause hematological change yielded similar OR in out-of-range analysis.** In this sensitivity analysis, information on medications prescribed during pregnancy (conception to 30 weeks' gestation) was acquired from the electronic health record, and a subset of the discovery cohort was developed of pregnancies without medications that could plausibly cause changes to blood cells (see S1 Appendix Text L for details on medications). Odds ratios, positive predictive values, and negative predictive values for each CBC index, outcome pair are listed in the table, as is the prevalence of each outcome. *P*-values are generated from Wald tests. Significance is denoted if $p < 0.0003$. *Abbreviations:* PPV, positive predictive value; HDP, hypertensive disorder of pregnancy; SGA, small for gestational age.
(XLSX)

**S14 Table. Pregnancies with rare longitudinal changes between 7−14 and 26−29 weeks' gestation are at higher risk of complications.** Assocations were tested in $N = 29{,}416$ after which those with preterm birth at <30 weeks' gestation or evidence of HDP or preeclampsia before 29 weeks were excluded ($N = 297$). Rare change thresholds are reported in the following units: HGB (g/dL), HCT (%), RBC ($10^6$/mm$^3$), PLT ($10^3$/μL), MCV (fL), WBC ($10^3$/μL), RDW (%), MCH (pg), MCHC (g/dL). In the table, we report the odds ratio, the number of pregnancies with the rare behavior (Total *N*), the number of complicated pregnancies with a rare behavior (Complicated *N*), the number of complicated pregnancies with a rare behavior that fall within the reference interval at 26−29 weeks' gestation (Complicated *N* in range), and their percentage and associated confidence interval. We also report the prevalence of the condition in the dataset, the positive predictive value (PPV), and the negative predictive value (NPV). *P*-values are generated from Wald tests. Threshold for statistical significance of the odds ratio was $p < 9*10^{-6}$ with Bonferroni correction. *Abbreviations:* HDP, hypertensive disorders of pregnancy; SGA, small for gestational Age; PPV, positive predictive value; NPV, negative predictive value; OR, odds ratio; CI, confidence interval;
(XLSX)

**S15 Table. Odds ratios and PPVs for rare behaviors are consistent in an out-of-sample validation cohort.** Out-of-sample validation of rare dynamics for hemoglobin HGB (g/dL), red blood cell count RBC ($10^6$/mm$^3$), hematocrit HCT (%),

platelet count PLT ($10^3$/µL), and mean red cell volume MCV (fL) shows odds ratios, PPV, and NPV consistent with the discovery cohort. Associations were tested in $N = 48,490$ pregnancies (with $N = 378$ preterm births at <30 weeks' gestation or pregnancies with evidence of HDP or preeclampsia before 29 week excluded). In the table, we report the odds ratio, the number of pregnancies with the rare behavior (Total $N$), the number of complicated pregnancies with a rare behavior (Complicated $N$), the number of complicated pregnancies with a rare behavior that fall within the reference interval at 26–29 weeks' gestation (Complicated $N$ in range), and their percentage and associated confidence interval. We also report the prevalence of the condition in the dataset, the positive predictive value (PPV), and the negative predictive value (NPV). We also show the prevalence of each adverse outcome and the associated confidence interval obtained via bootstrapping. *P*-values are generated from Wald tests. Threshold for statistical significance of the odds ratio was $p < 0.003$ with Bonferroni correction.
(XLSX)

**S16 Table. Sensitivity analysis of associations on an expanded discovery cohort including pregnancies ending in stillbirth and blood disorders previously excluded yielded near identical OR in rare behavior analysis.** In this sensivitiy analysis, the discovery cohort was re-derived to include pregnancies ending in stillbirth and blood disorders ($N = 29,775$). Pregnancies ending in stillbirth were labelled as having the composite outcome, and marked as preterm or HDP when applicable. The presence of a blood-related disorder was added as a binary covariate to our logistic regression. In the table, we report the odds ratio, the number of pregnancies with the rare behavior (Total $N$), the number of complicated pregnancies with a rare behavior (Complicated $N$), the number of complicated pregnancies with a rare behavior that fall within the reference interval at 26–29 weeks' gestation (Complicated $N$ in range), and their percentage and associated confidence interval. We also report the prevalence of the condition in the dataset, the positive predictive value (PPV), and the negative predictive value (NPV). We also show the prevalence of each adverse outcome and the associated confidence interval obtained via bootstrapping. Threshold for statistical significance of the odds ratio was $p < 0.0009$ with Bonferroni correction. *Abbreviations:* PPV, positive predictive value; HDP, hypertensive disorder of pregnancy; SGA, small for gestational age.
(XLSX)

**S17 Table. Sensitivity analysis of associations in the discovery cohort ($N = 34,159$) including the presence/absence of smoking, auto-immune disorders, and infections during pregnancy yielded near identical OR in rare behavior analysis.** In this sensitivity analysis, information on infections during pregnancy, pre-existing auto-immune disorders, and smoking status was acquired from the electronic health record, and binary covariates were developed reflecting their presence or absence (see S1 Appendix Text K for details. In the table, we report the odds ratio, the number of pregnancies with the rare behavior (Total $N$), the number of complicated pregnancies with a rare behavior (Complicated $N$), the number of complicated pregnancies with a rare behavior that fall within the reference interval at 26–29 weeks' gestation (Complicated $N$ in range), and their percentage and associated confidence interval. We also report the prevalence of the condition in the dataset, the positive predictive value (PPV), and the negative predictive value (NPV). We also show the prevalence of each adverse outcome and the associated confidence interval obtained via bootstrapping. *P*-values are generated from Wald tests. Threshold for statistical significance of the odds ratio was $p < 0.0009$ with Bonferroni correction. *Abbreviations:* PPV, positive predictive value; HDP, hypertensive disorder of pregnancy; SGA, small for gestational age.
(XLSX)

**S18 Table. Sensitivity analysis of associations in a subset of the discovery cohort ($N = 5,764$) with no medications that may cause hematological change yielded similar OR in rare behavior analysis.** In this sensitivity analysis, information on medications prescribed during pregnancy (conception to 30 weeks' gestation) was acquired from the electronic health record, and a subset of the discovery cohort was developed of pregnancies without medications that

could plausibly cause changes to blood cells (see S1 Appendix Text L for details on medications). In the table, we report the odds ratio, the number of pregnancies with the rare behavior (Total *N*), the number of complicated pregnancies with a rare behavior (Complicated *N*), the number of complicated pregnancies with a rare behavior that fall within the reference interval at 26–29 weeks' gestation (Complicated *N* in range), and their percentage and associated confidence interval. We also report the prevalence of the condition in the dataset, the positive predictive value (PPV), and the negative predictive value (NPV). We also show the prevalence of each adverse outcome and the associated confidence interval obtained via bootstrapping. *P*-values are generated from Wald tests. Threshold for statistical significance of the odds ratio $p < 0.0009$ with Bonferroni correction. *Abbreviations:* PPV, positive predictive value; HDP, hypertensive disorder of pregnancy; SGA, small for gestational age.
(XLSX)

**S19 Table. Sensitivity analysis of anemia in rare dynamics.** We considered the clinical definition of anemia of hemoglobin <11 g/dL. N(%) denotes the number of individuals with or without anemia and have rare dynamics. The percentage is calculated on the total of individuals with anemia or in those without anemia, to reflect an increased prevalence of rare dynamics in pregnancies with clinical anemia. For each group, we then evaluated the capability of detecting complications and reported the number of pregnancies with the Composite outcome, the sensitivity, and the positive predictive value (PPV).
(XLSX)

## Author contributions

**Conceptualization:** Veronica Tozzo, Rachel Petherbridge, Kaitlyn James, John M. Higgins, Camille E. Powe.

**Data curation:** Veronica Tozzo, Rachel Petherbridge, Kaitlyn James, Sarah Hsu, Deepti Pant, Brody H. Foy, Christopher Mow, Carolina Batlle Camero.

**Formal analysis:** Veronica Tozzo, Rachel Petherbridge, Tanayott Thaweethai, John M. Higgins, Camille E. Powe.

**Funding acquisition:** John M. Higgins, Camille E. Powe.

**Investigation:** Veronica Tozzo, Rachel Petherbridge, John M. Higgins, Camille E. Powe.

**Methodology:** Veronica Tozzo, Rachel Petherbridge, Tanayott Thaweethai, John M. Higgins, Camille E. Powe.

**Resources:** Christopher Mow, John M. Higgins, Camille E. Powe.

**Software:** Christopher Mow.

**Supervision:** John M. Higgins, Camille E. Powe.

**Validation:** Veronica Tozzo, Rachel Petherbridge, John M. Higgins, Camille E. Powe.

**Visualization:** Veronica Tozzo, Rachel Petherbridge, John M. Higgins, Camille E. Powe.

**Writing – original draft:** Veronica Tozzo, Rachel Petherbridge, John M. Higgins, Camille E. Powe.

**Writing – review & editing:** Veronica Tozzo, Rachel Petherbridge, Kaitlyn James, Sarah Hsu, Chloe Michalopoulos, Brody H. Foy, Tanayott Thaweethai, Christopher Mow, Jacqueline Maya, Carolina Batlle Camero, Lydia Shook, Kathryn J. Gray, Logan Mauney, John M. Higgins, Camille E. Powe.

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
