## [Editor Report · Decision Letter 0]

8 Jul 2025

Dear Dr Tozzo,

Thank you for submitting your manuscript entitled "Associations between hematologic dynamics during pregnancy and obstetric complications" for consideration by PLOS Medicine.

Your manuscript has now been evaluated by the PLOS Medicine editorial staff as well as by an academic editor with relevant expertise and I am writing to let you know that we would like to send your submission out for external peer review.

For clinical studies, please upload a copy of your trial study protocol as a supporting information file. The study protocol should be the version submitted for approval to the institutional review board or ethics committee, should include any amendments to the study protocol, as well as the date of their approval by the institutional review or ethics committee. Please also detail any deviations from the study protocol in the Methods section of your manuscript. The editors will consider the protocol and study conduct prior to a final decision for external review.

Please re-submit your manuscript within two working days, i.e. by Jul 10 2025 11:59PM.

Kind regards,

Louise Gaynor-Brook, MBBS PhD

Senior Editor

PLOS Medicine

---

## [Decision Letter · Decision Letter 1]

3 Nov 2025

Dear Dr Tozzo,

Many thanks for submitting your manuscript "Associations between hematologic dynamics during pregnancy and obstetric complications" (PMEDICINE-D-25-02420R1) to PLOS Medicine. The paper has been reviewed by subject experts and a statistician; their comments are included below and can also be accessed here: [LINK]

As you will see, the reviewers have shown interest in your study, but also raise several concerns. After discussing the paper with the editorial team and an academic editor with relevant expertise, I'm pleased to invite you to revise the paper in response to the reviewers' comments. We ask you to address all concerns, but specifically the concerns regarding remaining confounding, and risk of selection bias. We plan to send the revised paper to some or all of the original reviewers, and we cannot provide any guarantees at this stage regarding publication.

We ask that you submit your revision by Nov 24 2025 11:59PM. However, if this deadline is not feasible, please contact me by email, and we can discuss a suitable alternative.

Don't hesitate to contact me directly with any questions (sbruijn@plos.org).

Best regards,

Suzanne

Suzanne De Bruijn, PhD

Associate Editor

PLOS Medicine

sbruijn@plos.org

Comments from the academic editor:

There seems to be a high risk of selection bias, which is maximized by including only patients who delivered after 30 weeks, and by only including live births.

Comments from the reviewers:

Reviewer #1: The statistical methodology appears sound, and I acknowledge that this is not an area of my expertise.

The purpose of the study was to evaluate the association between hematologic dynamics and pregnancy complications. The authors reported that elevated red cell measures correlated

with subsequent obstetric complications. There were no such trend seen with platelets (which is surprising as low platelets are an identified finding in a subset of HDP--HELLP).

The authors excluded patients who delivered <30w for this study, and this may have excluded those with HDP resulting in preterm delivery, and perhaps those with HELLP.

Can the authors comment on how they manage CBCs done at 26-29w if those with evolving HDP may have multiple tests drawn at that time--i.e which CBC was used.

It is known previously (as reference in the discussion) that hemoconcentration can be seen in patients with HDP. In clinical practice, many signs are considered in the diagnosis of HDP. These will include, as the authors are aware blood pressure readings, liver function tests , renal function tests, in addition to platelet counts etc. Although clinicians are aware that hemoconcentration can also be seen, this variable might not be used often in clinical decision making. Patients with ongoing anemia (some studies estimate that as many as 40% women are in pregnancy) might not "trigger" a clinician's suspicion of HDP if the Hb "normalizes". The application of this finding into clinical setting might be limited.

The authors mentioned that they excluded those with anemia in sensitivity analysis--how did the authors define anemia?

Were the authors able to elicit the use of any Fe supplementation (oral or intravenous) in this discovery or validation cohort?

The manuscript was well-written.

Reviewer #2: Thank you for the invitation to review this manuscript by Tozzo et al which aims to determine reference ranges for CBC in pregnancy and assesses how change during pregnancy is associated with perinatal outcomes. The team have utilised a large cohort and two distinct time periods to allow for discovery and validation which is a strength to the study. However, there are some concerns with the study:

Major points:

1. Overall, the results are difficult to follow, with reference ranges, many exposures, times points and outcomes. The authors should consider reducing exposures and sub-analyses. Although a correction for multiple testing has been applied type-1 error is still possible.

2. A major consideration is how adjusted models were selected. There are likely other factors that are confounders that should be considered - autoimmune disease, infections, maternal medication use, and smoking, which have not been considered or adjusted for. The potential for residual confounding is high and greatly undermines the interpretation of results. Additionally, how were blood disorders handled in the validation cohort? These were excluded in the discovery cohort but not validation.

3. Why do inclusion/exclusion criteria differ between the validation and discovery cohorts? This needs further justification.

4. Projected models may introduce significant bias and do not form the main analysis, the author should consider removing this.

5. It is unclear how 'rare' changes were defined and must be updated. Is it possible that these rare values are outliers/errors in ascertainment/analysis?

6. How was missing data handled? It's unclear from the tables what proportion of missing data was in the dataset. What does extrapolated based on population model mean for missing BMI? This is not how missing data is commonly handled and is confusing given the authors have also stated that when BMI was left as missing if not recorded (legend for Table 1).

7. The authors have not included the number in each group/analysis, this should be added throughout. Given <10% had all time points it is likely the numbers are low for several analyses.

8. How were changes to diagnostic criteria for PE handled over the study period and different time points?

9. Reference intervals are not described in the results

Minor points:

1. Abstract is lacking results on defining reference ranges

2. The introduction could be expanded to describe the longitudinal changes and associations found outside of pregnancy

3. Is the third time point simply >36 weeks? (described in methods as up to 7 days before 37 weeks).

4. Materna clustering is described in the supplement but details of this should be included in methods

5. Results of validation should be discussed and included in the main text and described how they are similar or differ to discovery

6. Figure 3 is difficult to understand. The discovery and validation OR and 95% CI should be presented individually as per the methods.

Reviewer #3: this is a well-written manuscript with robust statistics

I have no recommendations for revisions

Reviewer #4: This is a valuable contribution on the haematological changes associated with pregnancy.

The normal range data largely confirms the comparative reference for Hb and Hct, with some greater discrepancies for some of the RBC indices, presumably due to the different methodologies employed.

The relatively lower value of diminished/diminishing haemoglobin/haematocrit values in pregnancy compared to increasing values is perhaps surprising and deserves a little bit more of discussion.

MAJOR POINTS:

The use of three different haematology analyzers and two if not three different technologies is a substantial limitation of this work and may explain the differences reported here for some of the red cell parameters compared with other data sets. It is important to convince the readers that this methodological heterogeneity is not causing some of your findings. Can you provide sub-analysis which can address these concerns? For example is your 2016-2022 Sysmex dataset different for some of the RBC parameters with the Advia one ontained prior to 2012? Same for the Sysmex He-5000 vs Sysmex HN-9000, since some of the RBC technology differs in these systems.

Minor points:

Please discuss how your CBC methodologies compare with the comparative reference ( Maternal Health Cohort ). Please discuss if there are notable differences between the two data set in methodologies.

-I could not see any reference to ethnicity and age for the studies cohort.

- It would be valuable to have more clinical data on the pregnancies with complications, especially blood pressure values and how they may relate to changes in hematological parameters. Same for renal function and at least proteinuria. The interpretation of the significance of the noted hematological changes is a challange in the absence of more granular clinical data outside of the outcome.

- I would like to have a more in depth discussion of PLT changes, especially if they appear int he absence or presence of other CBC changes. What are the characteristics of patients with decreasing PLT count and their susceptibility t o complications of pregnancy?

---

* Please upload any figures associated with your paper as individual TIF or EPS files with 300dpi resolution at resubmission; please read our figure guidelines for more information on our requirements: http://journals.plos.org/plosmedicine/s/figures. While revising your submission, we strongly recommend that you use PLOS's NAAS tool (https://ngplosjournals.pagemajik.ai/artanalysis) to test your figure files. NAAS can convert your figure files to the TIFF file type and meet basic requirements (such as print size, resolution), or provide you with a report on issues that do not meet our requirements and that NAAS cannot fix.

After uploading your figures to PLOS's NAAS tool - https://ngplosjournals.pagemajik.ai/artanalysis, NAAS will process the files provided and display the results in the "Uploaded Files" section of the page as the processing is complete.

If the uploaded figures meet our requirements (or NAAS is able to fix the files to meet our requirements), the figure will be marked as "fixed" above. If NAAS is unable to fix the files, a red "failed" label will appear above.

When NAAS has confirmed that the figure files meet our requirements, please download the file via the download option, and include these NAAS processed figure files when submitting your revised manuscript.

* Thank you for including a data availability statement. As these data cannot be made public, please include details on how researchers could gain access to these data.

* Please include the IRB approval number in your ethics statement.

* Financial disclosure: Please include URLs to the funders.

FIGURES AND TABLES

SUPPLEMENTARY MATERIAL

REFERENCES

OBSERVATIONAL STUDIES

* Abstract: Please include the study design, population and setting, number of participants, years during which the study took place (enrollment and follow up), length of follow up, and main outcome measures.

* [FOR POPULATION HEALTH/REGISTRY STUDIES] Please ensure that the study is reported according to the RECORD guideline (available from https://www.record-statement.org) and include the completed checklist as Supporting Information. Please add the following statement, or similar, to the Methods: "This study is reported as per the Reporting of Studies Conducted using Observational Routinely-Collected Data (RECORD) guideline (S1 Checklist)." When completing the checklist, please use section and paragraph numbers, rather than page numbers.

* For all observational studies, in the manuscript text, please indicate: (1) the specific hypotheses you intended to test, (2) the analytical methods by which you planned to test them, (3) the analyses you actually performed, and (4) when reported analyses differ from those that were planned, transparent explanations for differences that affect the reliability of the study's results. If a reported analysis was performed based on an interesting but unanticipated pattern in the data, please be clear that the analysis was data driven.

* Please state in the Methods section whether the study had a prospective protocol or analysis plan. If a prospective analysis plan (from your funding proposal, IRB or other ethics committee submission, study protocol, or other planning document written before analyzing the data) was used in designing the study, please include the relevant document(s) with your revised manuscript as a Supporting Information file to be published alongside your study and cite it in the Methods section. A legend for this file should be included at the end of your manuscript. If no such document exists, please make sure that the Methods section transparently describes when analyses were planned, and when/why any data-driven changes to analyses took place. Changes in the analysis, including those made in response to peer review comments, should be identified as such in the Methods section of the paper, with rationale.

---

## [Decision Letter · Decision Letter 2]

27 Mar 2026

Dear Dr. Tozzo,

Thank you very much for re-submitting your manuscript "Associations between hematologic dynamics during pregnancy and obstetric complications" (PMEDICINE-D-25-02420R2) for review by PLOS Medicine.

I have discussed the paper with my colleagues and the academic editor and it was also seen again by two reviewers. I am pleased to say that provided the remaining editorial and production issues are dealt with we are planning to accept the paper for publication in the journal.

The remaining issues that need to be addressed are listed at the end of this email. Please ensure you address all points; In your rebuttal letter you should indicate your response to the reviewers' and editors' comments and the changes you have made in the manuscript. Please submit a clean version of the paper as the main article file. A version with changes marked must also be uploaded as a marked up manuscript file.

We look forward to receiving the revised manuscript by Apr 03 2026 11:59PM.

Sincerely,

Suzanne De Bruijn, PhD

Associate Editor

PLOS Medicine

plosmedicine.org

Requests from Editors:

GENERAL

* Please confirm that your title complies with PLOS Medicine's style. Your title must be nondeclarative and not a question. It should begin with main concept if possible. "Effect of" should be used only if causality can be inferred, i.e., for an RCT. Please place the study design ("A randomized controlled trial," "A retrospective study," "A modelling study," etc.) in the subtitle (ie, after a colon).

* Please change your title to: “Associations between hematologic dynamics during pregnancy and obstetric complications: a retrospective observational study”

* Please ensure that all abbreviations are defined at first use throughout the text.

*Consider using the full names of some of the indices in the results, and/or the discussion, for ease of reading for non-experts. I do appreciate/acknowledge they are already defined in the methods.

* Please confirm that all numbers presented in the abstract are present and identical to numbers presented in the main manuscript text.

* Competing interests: Thank you for including your competing interests. Please include a sentence "All other authors have no competing interests to declare".

*Funding: Please provide URLs for all funders. Please also ensure it is clear that the T32 grant is an NIH grant.

* Ethics statement. Thank you for including your ethics statement.

-Please use the full name of the MBG institutional review board, rather than the abbreviation.

-Please ensure that the ethics statement is correctly placed/referenced. In the meta-data, it is stated that the ethics statement is included in the methods, whereas in reality it can be found after the discussion in the manuscript.

* Regarding the comment from Reviewer 4, comment 3: We think the response to this comment in your revision is adequate. However, please include a sentence in the methods, that you conducted a systematic evaluation of parameter stability.

DATA AVAILABILITY

* PLOS Medicine requires that the de-identified data underlying the specific results in a published article be made available, without restrictions on access, in a public repository or as Supporting Information at the time of article publication, provided it is legal and ethical to do so. Please see the policy at http://journals.plos.org/plosmedicine/s/data-availability and FAQs at http://journals.plos.org/plosmedicine/s/data-availability#loc-faqs-for-data-policy

* PLOS defines the “minimal data set” to consist of the data set used to reach the conclusions drawn in the manuscript with related metadata and methods, and any additional data required to replicate the reported study findings in their entirety. Authors do not need to submit their entire data set, or the raw data collected during an investigation. Please submit the following data:

-The values behind the means, standard deviations and other measures reported;

-The values used to build graphs;

-The points extracted from images for analysis."

* Please include contact information regarding where the data can be obtained, and note this cannot be the study author.

* Please clarify whether the aggregated data can also not be shared legally, or include them in the manuscript as a supporting file.

* Figure 3: please include a statement in the legend that the numerical values can be found in table S3.

ABSTRACT

* Please confirm that your abstract complies with our requirements, including format (three sections: Background, Methods and Findings, and Conclusions) and providing all the information relevant to this study type https://journals.plos.org/plosmedicine/s/submission-guidelines#loc-abstract

* Abstract: Please consider whether any of gestational-age-specific reference intervals, specifically the ones that differ from the literature, would be worth including in the abstract. We mean the numerical values here, not just a description.

* Please mention the specific data source (e.g. : EHR-based Maternal Health Cohort; Massachusetts General Hospital) rather than the current description "from electronic health records at an academic medical center and affiliated health system in the United States". If this is not possible, please provide an explanation why.

AUTHOR SUMMARY

* In the author summary, in the final bullet point of 'What Do These Findings Mean?', please include the main limitations of the study in non-technical language.

FIGURES AND TABLES

* Please consider avoiding the use of red and green in order to make your figure more accessible. This is for Figure 1, Figure 3, as well as figures S3 and S7.

* When a p value is given, please specify the statistical test used to determine it in the legend. Please check this for all figures and tables, but specifically table S2, S4, S6-12.

* Please provide titles and legends for all figures and tables (including those in Supporting Information files). It is currently not completely clear what is the title, and what is the legend for each figure/table.

Comments from Reviewers:

Reviewer #1: The revisions have made the manuscript better.

The limitations would be the generalizability of this analysis to the population of pregnant people with preterm HDP requiring delivery (i.e <28w).

Reviewer #2: I thank the authors for their considered update of the manuscript, they have responded to all reviewers well.

---

## [Editor Report · Decision Letter 3]

20 Apr 2026

Dear Dr Tozzo,

On behalf of my colleagues and the Academic Editor, Isabelle Dehaene, I am pleased to inform you that we have agreed to publish your manuscript "Associations between hematologic dynamics during pregnancy and obstetric complications: a retrospective observational study" (PMEDICINE-D-25-02420R3) in PLOS Medicine.

PRESS

Sincerely,

Suzanne De Bruijn, PhD

Associate Editor

PLOS Medicine